∂ | Open Peer Review | Computational Biology | Research Article

# rRNA operon improves species-level classification of bacteria and microbial community analysis compared to 16S rRNA

Sohyoung Won,[1,2] Seoae Cho,[2] Heebal Kim[1,2,3]

**ABSTRACT** Precise identification of species is fundamental in microbial genomics and is crucial for understanding the microbial communities. While the 16S rRNA gene, particularly its V3-V4 regions, has been extensively employed for microbial identification, however has limitations in achieving species-level resolution. Advancements in long-read sequencing technologies have highlighted the rRNA operon as a more accurate marker for microbial classification and analysis than the 16S rRNA gene. This study aims to compare the accuracy of species classification and microbial community analysis using the rRNA operon versus the 16S rRNA gene. We evaluated the species classification accuracy of the rRNA operon,16S rRNA gene, and 16S rRNA V3-V4 regions using a BLAST-based method and a k-mer matching-based method with public data available from NCBI. We further performed simulations to model microbial community analysis. We accessed the performance using each marker in community composition estimation and differential abundance analysis. Our findings demonstrate that the rRNA operon offers an advantage over the 16S rRNA gene and its V3-V4 regions for species-level classification within the genus. When applied to microbial community analysis, the rRNA operon enables a more accurate determination of composition. Using the rRNA operon yielded more reliable results in differential abundance analysis as well.

**IMPORTANCE** We quantitatively demonstrated that the rRNA operon outperformed the 16S rRNA and its V3-V4 regions in accuracy for both individual species identification and species-level microbial community analysis. Our findings can provide guidelines for selecting appropriate markers in the field of microbial research.

**KEYWORDS** rRNA operon, 16S rRNA, bacteria, species, classification, microbial community, compositions, accuracy, identification, microbiome, ribosomal RNA, metagenomics

Accurate taxonomic classification is crucial for reliable outcomes in microbial genomics research. As analysis increasingly shifts toward species-level identification beyond the genus level, enhancing the resolution of microbial identification becomes critical for discerning specific species (1). This plays a significant role in discovering novel microbial species and fostering a comprehensive understanding of microbial communities (1).

Next-generation sequencing technologies have revolutionized microbial genomics, enabling rapid and cost-effective sequencing of whole genomes and amplicons (2). Second-generation sequencing platforms, like Illumina's HiSeq and MiSeq, generate millions of short reads (100–300 bp) (3), while third-generation technologies, like PacBio's SMRT and Oxford Nanopore's MinION, produce significantly longer reads (up to 100 kb or more) (4, 5).

16S rRNA gene sequencing is a widely used method for microbial identification and community profiling (6, 7). It targets the highly conserved 16S rRNA gene, containing

Address correspondence to Heebal Kim, heebal@snu.ac.kr.

The authors declare no conflict of interest.

variable regions among species. Some second-generation sequencing approaches using only specific variable regions (e.g., V3 and V4) as markers can be cost-effective but have limitations in taxonomic resolution (8). Even utilizing the entire 16S rRNA gene, accurate species-level classification remains challenging, potentially underestimating diversity and hindering accurate characterization of microbial communities (9, 10).

The emergence of third-generation sequencing technologies has enabled the analysis of larger genomic regions, paving the way for whole rRNA operon sequencing as a prominent approach (11). Encompassing the 16S, 23S, and 5S rRNA genes, along with the internal transcribed spacer regions, the rRNA operon provides a comprehensive framework for microbial identification and phylogenetic studies (12). Compared to 16S rRNA sequencing, rRNA operon sequencing offers richer information content, promising higher-resolution taxonomic classification, reaching the species level and more accurate microbiome community analysis (13). However, further quantitative research is required to fully validate these expectations. There is a study comparing the performance of phylogeny analysis using the rRNA operon versus the 16S, 23S, and 5S rRNA genes, but the accuracy of species classification was not assessed (14).

This study utilizes public data to compare the accuracy of species classification within the same genus using the entire rRNA operon sequence, the 16S, 23S, and 5S rRNA gene sequences, and the V3 and V4 regions of the 16S rRNA. Additionally, we create simulated microbiome community data to compare how accurately each region determines the proportion of each species. The aim is to provide guidelines for selecting marker regions for bacterial species classification and species-level microbiome studies.

## RESULTS

### Species classification accuracy within the genus

Both BLAST and k-mer matching methods demonstrated significantly higher accuracy, precision, and sensitivity when utilizing the entire rRNA operon compared to the 16S rRNA alone (Fig. 1; Table S2). The average accuracy for BLAST-based classification using the rRNA operon reached 0.999, with a standard deviation of 0.005. This accuracy dropped to 0.937 with a standard deviation of 0.109 when using the 16S rRNA, and further decreased to 0.702 with a standard deviation of 0.301 with the 16S rRNA V3-V4 regions. When using 23S rRNA, the average accuracy was 0.985, which was higher than that of 16S rRNA but lower than using the entire rRNA operon. The standard deviation of accuracy was 0.048, similarly falling between the values for 16S rRNA and the rRNA operon. For 5S rRNA, the average accuracy was the lowest at 0.500, and the standard deviation was the highest at 0.341. This trend reflects that analyzing broader genomic regions leads to improved accuracy and reduced variability.

k-mer matching yielded comparable results. The average accuracy using the rRNA operon was 0.999, exceeding the 0.919 observed for 16S rRNA and 0.706 for the V3-V4 regions. For 23S rRNA, the average accuracy was 0.975, and for 5S rRNA, it was 0.468. Thus, the accuracy ranked as follows: rRNA operon, 23S rRNA, 16S rRNA, 16S V3-V4 regions, and 5S rRNA. The rRNA operon also displayed the lowest standard deviation (0.006), compared to 0.063 for 23S rRNA, 0.124 for 16S rRNA, 0.293 for the V3-V4 regions, and 0.318 for 5S rRNA.

Precision and sensitivity trends mirrored those of accuracy. Using the rRNA operon, BLAST and k-mer matching achieved the highest average precision values, at 0.999 and 0.998, respectively. However, precision decreased for both methods when using the 16S rRNA (0.933 for BLAST and 0.895 for k-mer matching), and further dropped for the 16S rRNA V3-V4 regions (0.702 and 0.686 for BLAST and k-mer matching, respectively). For 23S rRNA, the average precision using BLAST was 0.986 with a standard deviation of 0.034, while for k-mer matching, it was 0.969 with a standard deviation of 0.634. The 5S rRNA showed the lowest precision, with an average of 0.459 and a standard deviation of 0.291 for BLAST, and an average of 0.385 and a standard deviation of 0.268 for k-mer matching. Similarly, while both methods maintained high sensitivity with the rRNA

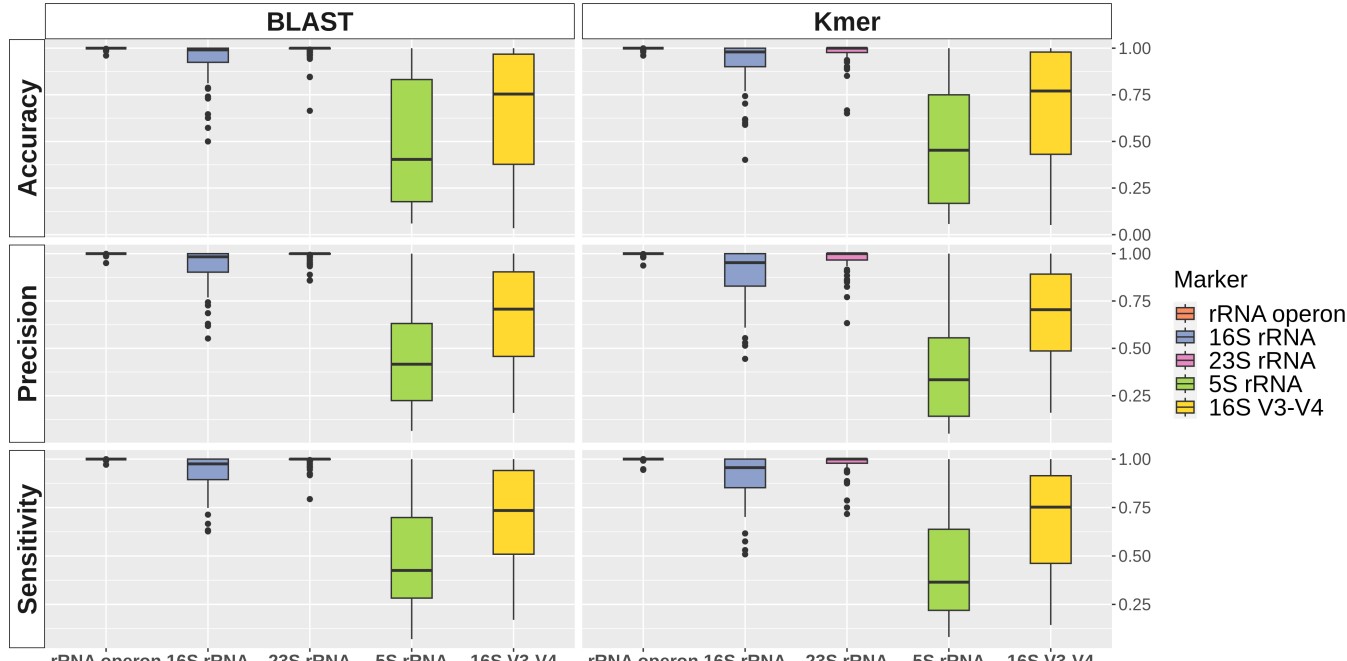

**FIG 1** A boxplot of species classification accuracy, precision, and sensitivity across genera using the rRNA operon versus the 16S rRNA gene, 23S rRNA gene, 5S rRNA gene, and the V3-V4 regions of 16S rRNA gene. The left panel demonstrates the results from the BLAST method, and the right panel presents outcomes from the k-mer matching method.

operon, above 0.998, sensitivity dropped for 16S rRNA, at 0.938 for BLAST and 0.910 for k-mer matching. For 23S rRNA, the average sensitivity was 0.989 for BLAST and 0.975 k-mer matching. The 5S rRNA again showed the lowest sensitivity, with an average of 0.503 for BLAST and an average of 0.433 for k-mer matching.

Across both methods, the *Haemophilus* genus exhibited the lowest accuracy with the rRNA operon, which was 0.960. For the 16S rRNA, the lowest accuracy was observed in the *Serratia* genus, with BLAST and k-mer matching methods reporting 0.500 and 0.598, respectively. *Listeria* displayed the lowest accuracy for 23S rRNA with both methods, 0.664 and 0.651 for BLAST and k-mer matching, respectively. Notably, employing the rRNA operon consistently achieved higher accuracy for all genera compared to either 23S rRNA or 16S rRNA, regardless of the chosen method (BLAST or k-mer matching). When using 23S rRNA, the accuracy was lower than when using 16S rRNA in the BLAST method for the *Acinetobacter*, *Bacillus*, *Lactococcus*, and *Listeria* genera, with differences ranging from 0.005 to 0.115. In k-mer matching, 23S rRNA had lower accuracy compared to 16S rRNA for the *Acinetobacter*, *Chlamydia*, *Lactococcus*, *Limosilactobacillus*, and *Stenotrophomonas* genera with differences ranging from 0.013 to 0.125.

On average, the rRNA operon resulted in 0.062 and 0.080 higher classification accuracy compared to 16S rRNA using BLAST and k-mer matching, respectively. Conversely, using 23S rRNA resulted in an average accuracy of 0.014 lower with BLAST and 0.023 lower with k-mer matching compared to the rRNA operon. The most significant difference in accuracy between the rRNA operon and 16S rRNA was observed in *Serratia* with a substantial difference of 0.500 (BLAST) and 0.598 (k-mer matching). Following *Serratia*, the genera most affected by the choice of marker (rRNA operon versus 16S rRNA) for accuracy in BLAST analysis were *Xanthomonas*, *Pseudomonas*, and *Rhizobium*, with differences of 0.424, 0.365, and 0.354, respectively. In k-mer matching, the subsequent largest accuracy disparities were seen for *Xanthomonas*, *Rhizobium*, and *Bacillus A*, with differences of 0.412, 0.385, and 0.380, respectively. The largest difference in accuracy between the rRNA operon and 23S rRNA was observed in the *Listeria* genus

(0.333 for BLAST and 0.335 for k-mer matching). The second largest difference was seen in *Bacillus A*, which was 0.155 for BLAST and 0.312 for k-mer matching.

Using the rRNA operon, the BLAST method achieved perfect species classification accuracy (1.0) in 92.8% (64) of genera, and the k-mer match method did so in 88.4% (61) of genera. In contrast, with the 16S rRNA, the BLAST method had less than 0.9 accuracy in 18.8% (13) of genera, and the k-mer matching method in 24.6% (17) of genera. This indicates that using the rRNA operon enables more precise species classification than the 16S rRNA. When using the 23S rRNA, 73.9% of the cases in BLAST and 59.4% in k-mer matching achieved an accuracy of 1. Additionally, there were three cases in BLAST and five cases in k-mer matching where the accuracy was below 0.9.

The standard deviation of accuracy was significantly higher with 16S rRNA compared to the rRNA operon. Specifically, it was 21.9 times higher with BLAST and 21.5 times higher with k-mer matching. Similarly, the standard deviation for 23S rRNA was also substantially higher than the rRNA operon, at 9.28 times higher with BLAST and 11.0 times higher with k-mer matching. In terms of minimum accuracy observed, the rRNA operon consistently outperformed both 16S rRNA and 23S rRNA. While the minimum accuracy for the rRNA operon always exceeded 0.95, the 16S rRNA gene could dip below 0.5 in some cases, and the 23S rRNA gene could sometimes fall below 0.7.

## Microbial community composition prediction

We conducted simulations to evaluate the effectiveness of different regions for predicting the species compositions (Fig. 2). These simulations assumed that species existed in random proportions following a Dirichlet distribution. The figure depicts the predicted proportions of the top 12 species for each method (rRNA operon, 16S rRNA, 23S rRNA, 5S rRNA, and 16S rRNA V3-V4 regions). Predictions using the rRNA operon closely matched the actual compositions. The 16S rRNA and 23S rRNA predictions displayed a similar trend to the actual compositions but with some discrepancies in the ratios between species. Predictions based on the 5S rRNA and 16S rRNA V3-V4 regions deviated significantly from the actual compositions.

To numerically verify these observed trends, we calculated the Pearson correlation coefficient between the actual and predicted proportions (Table 1). Across six

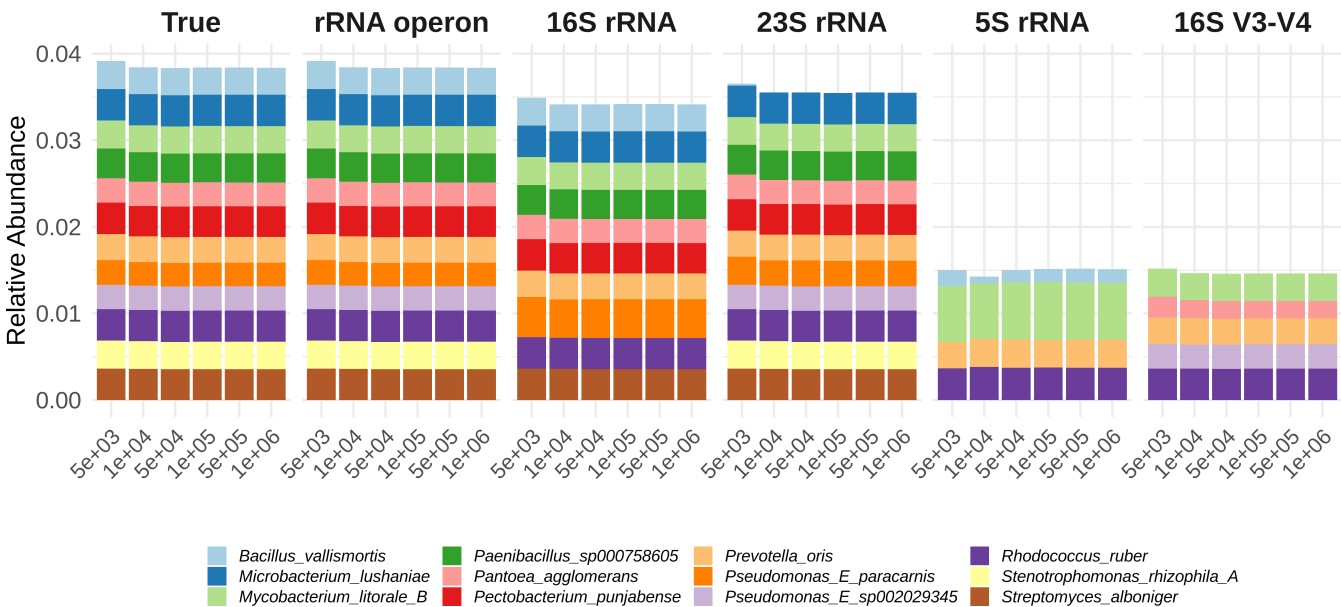

FIG 2 The relative abundance of the top 12 abundant species in the True data, where True represents the actual proportions. rRNA operon, 16S rRNA, 23S rRNA, 5S rRNA, and 16S V3-V4 show the proportions of species predicted based on the accuracy of species classification within those genomic regions. Each color represents the same species across different predictions, with the x-axis indicating the number of reads used in the simulation and the y-axis showing the proportion of each species.

**TABLE 1** Pearson correlations between actual species proportions and predicted species proportions using the rRNA operon, 16S rRNA, and 16S rRNA V3-V4 regions, by the number of reads used for the simulation

|  | 5,000 | 10,000 | 50,000 | 100,000 | 500,000 | 1,000,000 |
|---|---|---|---|---|---|---|
| rRNA operon | 0.998 | 0.998 | 0.998 | 0.998 | 0.998 | 0.998 |
| 16S rRNA | 0.804 | 0.821 | 0.835 | 0.837 | 0.838 | 0.838 |
| 16S V3-V4 | 0.295 | 0.321 | 0.344 | 0.346 | 0.348 | 0.348 |

simulations, the correlation between actual and predicted proportions using the rRNA operon remained consistently high, with an average of 0.999 and a standard deviation of 0.0001. This held true regardless of the number of reads used in the simulation. The 16S rRNA and 23S rRNA exhibited lower average correlations of 0.794 and 0.917, respectively, with higher standard deviations of 0.011 and 0.003, respectively. This indicates a poorer match to the actual proportions and greater variability between simulations compared to the rRNA operon. The 5S rRNA and 16S rRNA V3-V4 regions yielded even lower average correlations of 0.251 and 0.049, with standard deviations of 0.017 and 0.014, respectively. The correlation between actual and predicted proportions increased with the number of simulated reads, except for the rRNA operon, reaching a plateau beyond 500,000 reads.

To quantify the similarity between the actual and predicted microbial community compositions, we employed the Aitchison distance metric. A smaller Aitchison distance signifies greater similarity between the two datasets (Fig. 3A). The predicted composition using the rRNA operon yielded a remarkably close distance to the actual composition, averaging only 2.76 across six simulations. Conversely, the average distances observed when utilizing the 16S rRNA and 16S rRNA V3-V4 regions were higher at 23.2 and 42.6, respectively. The 23S rRNA displayed an intermediate average distance of 13.5, while the 5S rRNA showed the highest distance, 52.7. Predictions based on the rRNA operon exhibited the closest match to the actual community composition. Compared to 16S rRNA, the rRNA operon resulted in distances that were 8.47 times smaller. Similarly, it yielded distances 4.93 times lower than those obtained using the 23S rRNA.

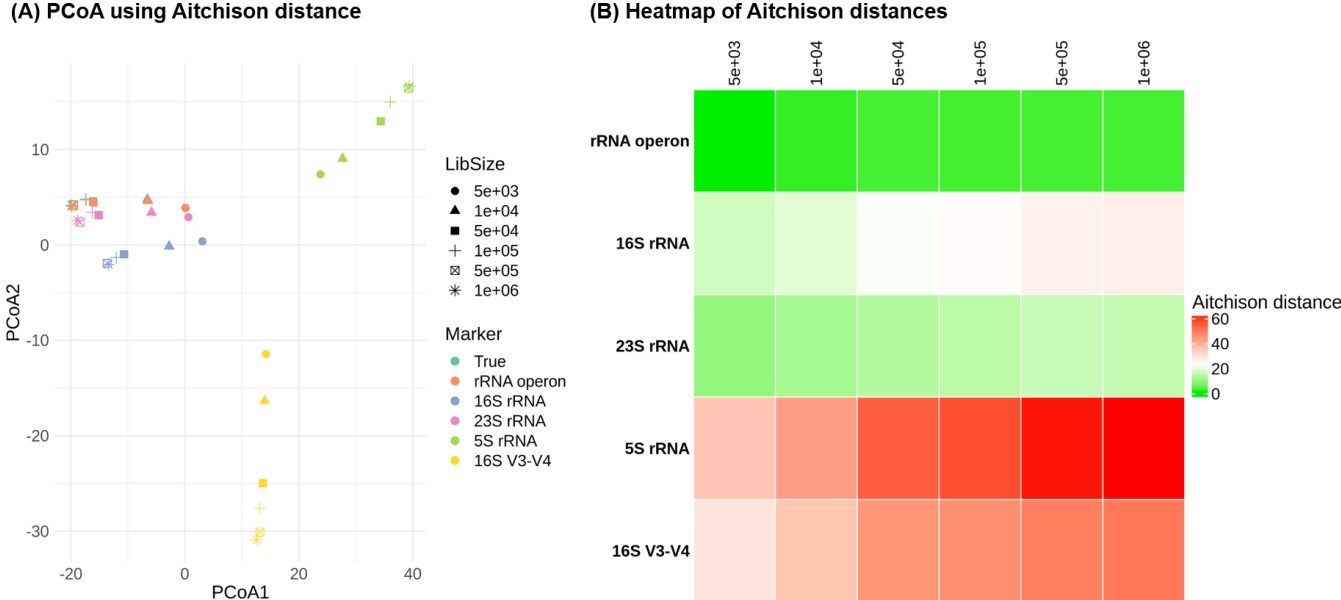

**FIG 3** Aitchison distances between true and predicted microbial community compositions. (A) Principal coordinate analysis (PCoA) plot using Aitchison distance. Closer points indicate greater similarity between the actual and predicted communities. (B) Heatmap of Aitchison distances between actual data and predicted data. Green indicates lower distances (greater similarity), while red indicates higher distances (lower similarity).

A principal coordinate analysis plot using Aitchison distance further visualized these findings (Fig. 3B). The predicted proportions based on the rRNA operon nearly perfectly overlapped with the actual data. The 23S rRNA data clustered next closest, followed by the 16S rRNA data. The 16S V3-V4 and 5S data points were substantially more distant. This reinforces the notion that smaller Aitchison distances correspond to greater similarity and more accurate simulation of the actual microbial community.

To statistically validate these observations, we conducted an analysis of similarities test using the Aitchison distance. This non-parametric method evaluates the probability of observed differences in similarity between groups arising by chance. The results confirmed these findings. Predictions made with the rRNA operon yielded a $P$ value of 0.214, indicating no statistically significant difference from the actual community composition. Conversely, predictions utilizing the 23S rRNA produced a $P$ value of 0.015, indicating a statistically significant difference at the $P < 0.05$ level. Predictions using the 16S rRNA, 5S rRNA, and 16S rRNA V3-V4 regions produced $P$ values of 0.003, 0.002, and 0.004, respectively, all indicating statistically significant differences at the $P < 0.01$ level. These significant $P$ values demonstrate that these methods yielded compositions statistically distinct from the actual community.

## Microbial community composition and differential abundance in human gut microbiome data

We evaluated the performance using the rRNA operon, 23S rRNA, 16S rRNA, 5S rRNA, and 16S rRNA V3-V4 regions for microbial community composition prediction using real human gut microbiome data (Fig. 4). For our analysis, we collected gut microbiome data from a study which investigated gut microbiomes of patients with Crohn's disease and healthy controls. We used the proportion data of 188 overlapping species from samples of 39 healthy controls and 47 patients with Crohn's disease. The correlation between the reference species proportions and those predicted using the rRNA operon was always 1.00, indicating perfect prediction. Predictions based on the 16S rRNA and the 16S V3-V4 regions exhibited lower correlations with the reference, with averages of 0.868 and 0.695, and standard deviations of 0.041 and 0.075, respectively. For the 23S rRNA, the average correlation between actual and predicted proportions was 0.945, with a

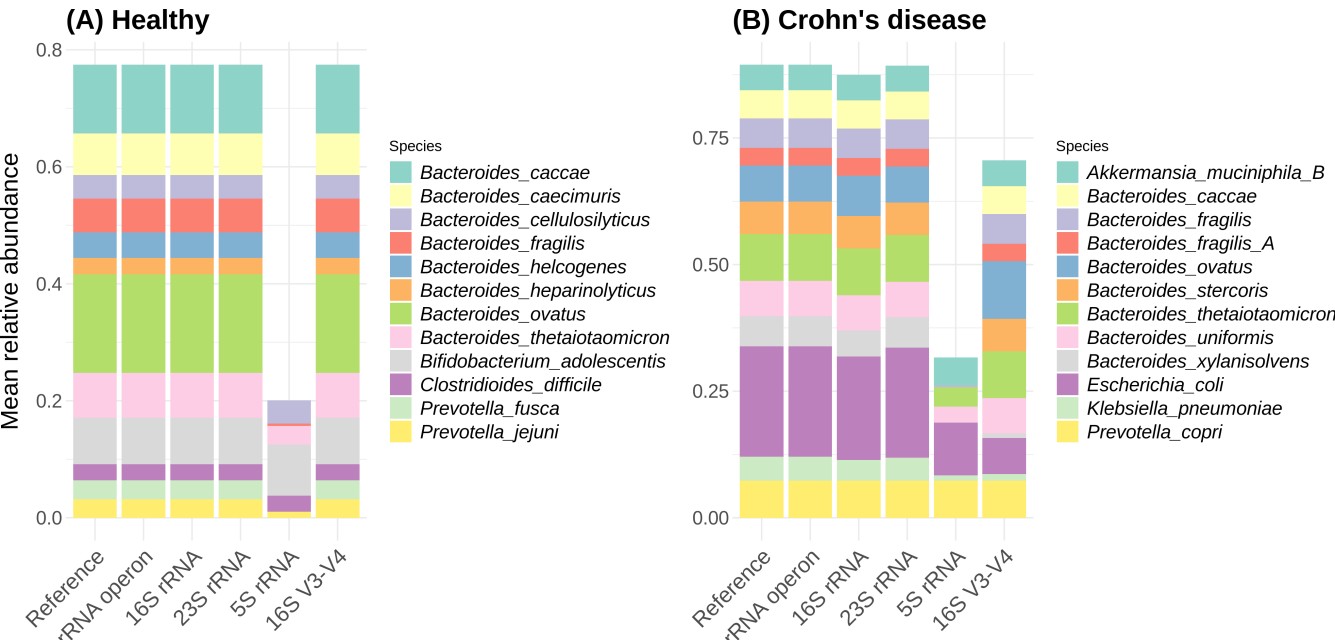

**FIG 4** The relative abundance of the top 10 gut microbiome species in (A) healthy individuals and (B) patients with Crohn's disease. Reference indicates actual proportions. rRNA operon, 16S rRNA, 23S rRNA, 5S rRNA, and 16S V3-V4 show predicted proportions based on species classification accuracy.

standard deviation of 0.025. The average was 0.388 and the standard deviation was 0.091 with the 5S rRNA. The rRNA operon consistently achieved high correlations between predicted proportions and reference proportions. Conversely, while the 23S rRNA had a high average correlation, the correlation dropped as low as 0.867 in one patient's data, indicating significant variability in prediction accuracy among samples. The 16S rRNA also exhibited a lower minimum correlation, dropping as low as 0.779. These results align with our observations from the randomly generated data. In addition, in the patient group, the 16S rRNA and 16S rRNA V3-V4 regions showed greater discrepancies in relative abundance analysis, with species such as *Escherichia coli* and *Bacteroides xylanisolvens* appearing less frequently compared to their actual proportions.

We further assessed the methods by conducting differential abundance analyses based on both the reference compositions and those predicted by each classification method. We compared species identified as significantly different in each case (Fig. 5). The reference data identified 20 significantly differentially abundant species, which were accurately reflected by the predictions made with the rRNA operon. The 16S rRNA identified 23 significant species, with 18 overlapping with the reference findings, while the 23S rRNA identified 21 significant species, of which 19 overlapped with the reference. The 16S rRNA missed two significant species (false negatives) while the 23S rRNA missed one significant species. Additionally, five species identified as significant by the 16S rRNA were not truly so (false positives), while the 23S rRNA produced two false positives. This translates to a false negative rate of 10.0% and a false discovery rate of 21.7% for the 16S rRNA results, and a false negative rate of 5.0% and a false discovery rate of 9.52%

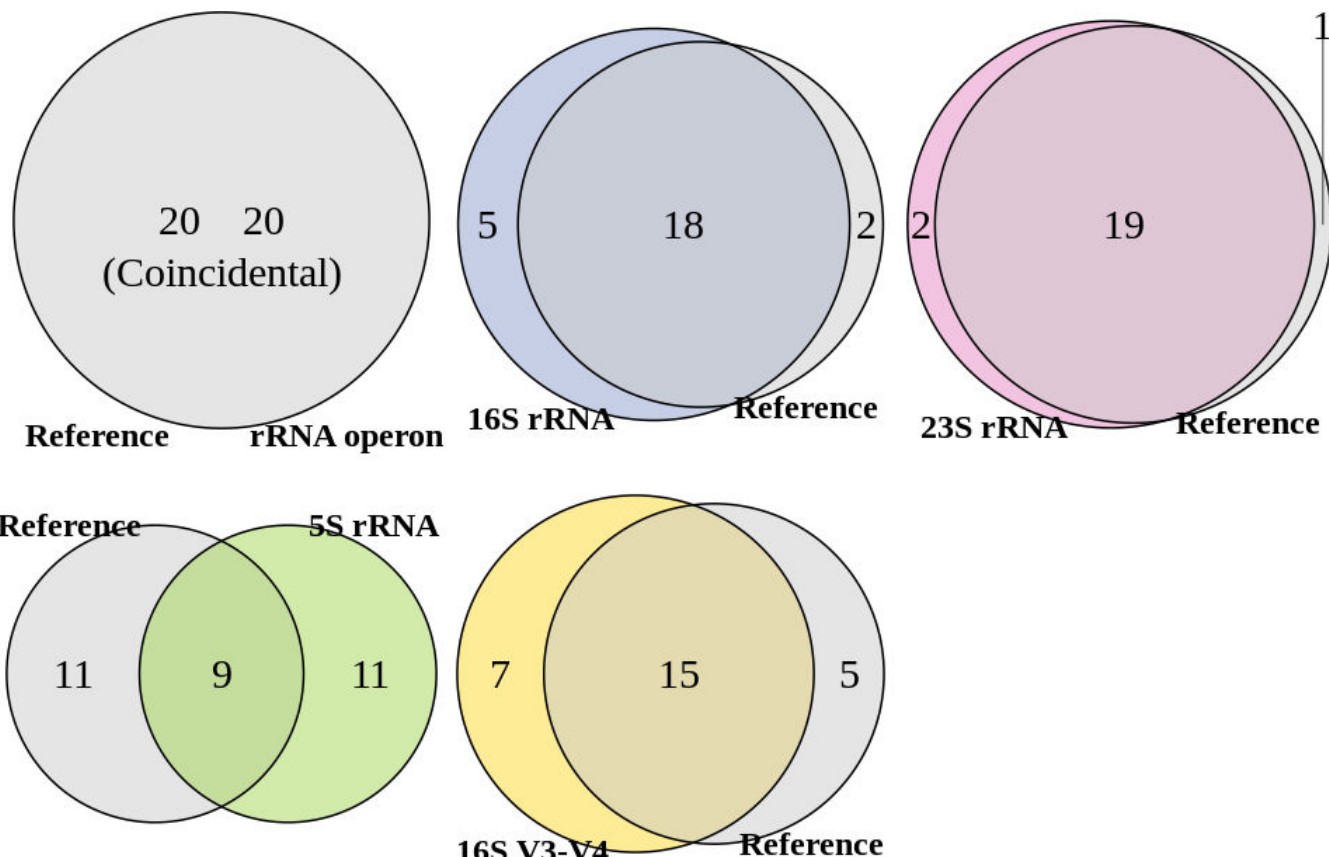

**FIG 5** Venn diagrams of the significant species identified through differential abundance analysis based on proportions derived from the reference and rRNA operon, reference and 16S rRNA, reference and 23S rRNA, reference and 5S rRNA, and reference and16S rRNA V3-V4 regions. Overlapping sections of the diagram represent the number of species significantly identified across both methods. The area exclusive to the Reference (left side) shows the number of species that were false negatives. Areas unique to each method (right side) indicate false positives. Complete overlap between the Reference and a method implies identical species significance findings.

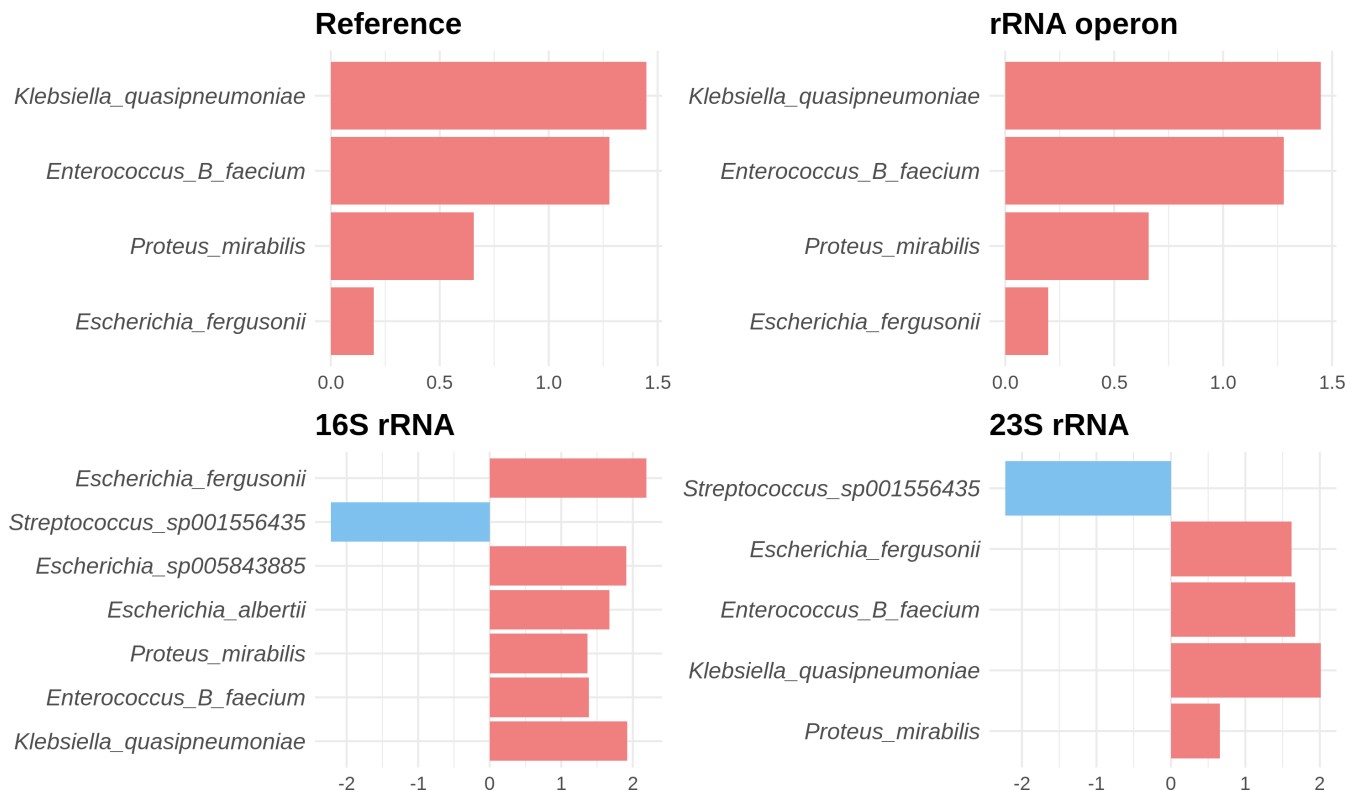

**FIG 6** The coefficients from differential abundance analyses using the proportions obtained from reference, rRNA operon, 16S rRNA, and 23S rRNA. We only showed species that have discrepancies in the reference and 16S rRNA results or in the reference and 23S rRNA; species identified as significant in one analysis but not the other and species with differing direction of coefficient. A positive coefficient (depicted in pink) indicates a species is more abundant in patients, while a negative coefficient (shown in sky blue) suggests it is less abundant. The magnitude of the coefficient signifies the degree of abundance difference.

for the 23S rRNA results. The 5S rRNA and 16S rRNA V3-V4 regions performed worse, with greater false negative rates (55.0% and 25.0%, respectively) and false discovery rates (55.0% and 31.8%, respectively).

Figure 6 depicts the coefficients of species that were identified as false negatives or false positives when using the 16S rRNA or the 23S rRNA. Here, the coefficient represents the relative abundance of a species in patients compared to donors. Some species identified as significantly different by 16S rRNA or 23S rRNA were not detected by the reference or the rRNA operon predictions. Additionally, there were cases where the magnitudes of the coefficients differed between the reference findings and those of the 16S rRNA, as well as between the reference findings and those of the 23S rRNA. Conversely, the results using the rRNA operon displayed a high degree of agreement with the reference data, with consistent signs and magnitudes of coefficients for all species.

## DISCUSSION

The rRNA operon demonstrably outperformed the 16S rRNA gene in terms of species classification accuracy. Statistical tests confirmed this observation. Paired Wilcoxon rank sum tests revealed highly significant differences ($P < 0.0001$ for BLAST and k-mer matching) in accuracy favoring the rRNA operon. Also, the accuracy of the rRNA operon was significantly higher in both methods compared to the 23S rRNA ($P < 0.001$). Furthermore, the rRNA operon exhibited considerably lower variability in accuracy across genera. This signifies that the rRNA operon offers consistently high and stable classification accuracy across various genera, while the 16S rRNA can yield unreliable results due to substantial variations in accuracy depending on the genus. This trend was also

evident in terms of precision and sensitivity. Accuracy in classification measures the proportion of all predictions (both positive and negative) that are correct. Precision, however, specifically assesses the accuracy of positive predictions, indicating the fraction of predicted positives that are truly positive. Sensitivity, or recall, evaluates how well the model identifies actual positives. Similar trends were observed across all three metrics, indicating that the rRNA operon is more reliable and effective for accurate species identification.

Additionally, both the 23S rRNA and 16S rRNA showed lower performance in species classification using k-mer matching compared to the BLAST method. A possible reason for this difference is that BLAST classifies species based on the overall similarity of sequences, while k-mer matching classifies species based on the exact match count of 31-mers. This could make k-mer matching less effective in identifying species when the sequences have high variability or when partial matches are important for classification.

Simulations revealed a clear advantage for the rRNA operon in predicting species compositions within microbial communities. The correlation between actual and predicted proportions using the rRNA operon consistently outperformed both the 16S rRNA and the 23S rRNA. Notably, the correlation with the 16S rRNA V3-V4 regions was significantly lower, rendering it unreliable for capturing meaningful relationships with the actual data. When using the rRNA operon versus the 16S rRNA, the difference in correlation between actual and predicted compositions in the data was greater than the difference in accuracy of individual species predictions. While the performance using 23S rRNA surpassed 16S rRNA, it remained inferior to the rRNA operon. This suggests that analyzing the entire rRNA operon offers a more effective approach compared to solely using the 23S rRNA gene.

The Aitchison distance metric further reinforces the superiority of the rRNA operon. Notably, only proportions predicted using the rRNA operon exhibited no statistically significant difference from the actual proportions in the distance analysis. In contrast, all other methods showed significant discrepancies between the actual and predicted proportions. These findings collectively suggest that the rRNA operon provides a more accurate reflection of the true microbial community structure compared to 16S rRNA or 23S rRNA alone.

Simulations replicating the composition of actual human gut microbiomes yielded consistent results. Notably, this lower accuracy in predicting microbial compositions using 16S rRNA, 23S rRNA, and the V3-V4 regions of 16S rRNA was particularly problematic for the patient group. In the patient group, the average correlation between predicted and reference compositions for the rRNA operon remained at 1.00, both the 16S rRNA and 23S rRNA showed lower correlations. Specifically, the 16S rRNA achieved an average correlation of 0.859, with 4 out of 47 patient samples showing a correlation below 0.8. Similarly, the 23S rRNA had two patient samples with a correlation below 0.8. In the 16S rRNA V3-V4 regions, the correlation in the patient group dropped to as low as 0.433.

The observed difference in accuracy for community composition predictions also impacted the results of differential abundance analyses. When utilizing proportions derived from the rRNA operon, the analysis identified the same 20 significant species as those identified using the reference proportions, indicating perfect agreement. In contrast, the analysis based on other markers yielded discrepancies, further solidifying the limitations of these methods for accurate prediction.

Using the rRNA operon as a marker provided higher accuracy in individual species classification than using the 16S rRNA or its V3-V4 regions, leading to more accurate community composition predictions and more reliable results in differential abundance analyses. This is due to the rRNA operon being longer and containing more information than the 16S rRNA or its V3-V4 regions. In our study, the average length of the rRNA operon was 5,127.13 bp, the 16S rRNA averaged 1,535.72 bp, and the V3-V4 regions averaged 399.45 bp. The 23S rRNA, with a length of 2,914.39 bp, was longer than the 16S rRNA but shorter than the rRNA operon, resulting in intermediate accuracy. The

5S rRNA, being the shortest at 108.29 bp, showed the lowest accuracy. This suggests that the length of the marker region is proportional to the accuracy of species classification and community composition prediction. However, sequencing costs may increase with the breadth of the region being read (15). Consequently, the choice of method should consider the required resolution and available budget. The 16S rRNA can be a suitable option when less precision is acceptable, species-level analysis is not necessary, and genus-level identification suffices. On the other hand, for research requiring precise species-level analysis, such as discovering biomarkers, utilizing the microbiome for treatments, or other studies necessitating accurate species identification, the rRNA operon is preferable. This is especially true for disease-related microbial community studies, as the accuracy difference in community composition predictions between methods was more pronounced in patient groups, highlighting the importance of using the rRNA operon for more precise species differentiation in such contexts. Using 23S rRNA allows for more accurate species-level analysis compared to 16S rRNA, but it also requires sequencing a broader region, similar to the rRNA operon. Moreover, the 23S rRNA showed limitations in the community analysis of patient data. There is also a study demonstrating that using the rRNA operon for phylogenetic analysis is more accurate than using individual rRNA genes, supporting our findings (14). Real-world experiments, such as species-level comparisons achieved by sequencing both the rRNA operon and other regions from the same sample, could further solidify our findings.

The accuracy of microbial community composition prediction using the 23S rRNA, 16S rRNA, or 16S rRNA V3-V4 regions improves when using more number of reads, which means sequencing more data. However, this also raises data production costs and should be carefully weighed. Given the same budget, producing less data with the rRNA operon may be more efficient than generating more data with other rRNA genes.

Not only accuracy but also feasibility should be considered when choosing a marker for microbial community analysis. For sequencing, the existence of a universal sequencing primer is necessary. Additionally, from an analytical standpoint, a well-established database and pipeline are crucial for deriving species abundance from sequencing data. The 16S rRNA gene and its variable regions have long been established as well-defined markers, with readily available primers and comprehensive databases (16). The rRNA operon is also feasible as a marker for species identification. Seol et al. (17) demonstrated the feasibility of species identification using the rRNA operon by developing the MIrROR database and analysis pipeline. This study also provides examples of universal primer sequences. There is an example of species-level analysis using the MIrROR database and pipeline (18).

Employing the rRNA operon as a marker demonstrably enhances individual species classification accuracy compared to other rRNA genes. This translates to more precise predictions of microbial community compositions and more reliable differential abundance analysis results. The 16S rRNA V3-V4 regions exhibited even lower accuracy across all scenarios compared to the full 16S rRNA, highlighting a significant decline in precision. Therefore, for research requiring accurate species classification, employing the rRNA operon as a marker appears to be the most appropriate choice. In microbial community studies aiming for precise species-level analysis, utilizing the rRNA operon is advisable as using the 16S rRNA has its limitations, and relying solely on its V3-V4 regions may make it challenging to achieve meaningful results. However, as our results are based on simulations, it would be beneficial to conduct actual sequencing experiments to confirm our findings.

## MATERIALS AND METHODS

### Data collection

We collected complete bacterial genomes available in the NCBI database as of 29 November 2023 (19). To ensure robust comparisons, we only considered genera containing more than 50 complete genomes. For quality control, we removed atypical

genomes and metagenome-assembled genomes. We assigned taxonomic classifications to our genomes using the Genome Taxonomy Database (GTDB) (20). This involved matching the accession IDs of our samples against the records in the GTDB. We then employed the corresponding species and genera identified through this matching process for our study. Our final data set comprised 69 genera, 1,926 species, and 18,976 genome sequences (Table S1).

## rRNA operon and 16S rRNA sequence extraction

We utilized riboSeed with its default settings to extract rRNA operon sequences (21). Following identification of rRNA gene regions using the riboscan command, we employed the riboselect command to locate rRNA operon regions containing 16S, 23S, and 5S rRNA. The corresponding rRNA operon sequences were then extracted. Implementing quality control, only sequences within the 4,000–6,000 bp range were retained. 16S, 23S, and 5S rRNA sequences were extracted from regions identified as 16S, 23S, and 5S rRNA by the riboscan results.

We employed EMBOSS's primer search tool to identify the V3-V4 regions within the previously extracted 16S rRNA sequences (22). The primer sequences used were CCTACGGGNGGCWGCAG for the forward primer and GACTACHVGGGTATCTAATCC for the reverse primer (23). A mismatch percentage of 10% was allowed during the search. Following quality control, only sequences with lengths between 430 bp and 550 bp were retained.

## Introduction of sequencing errors

To simulate real-world applications of using 16S rRNA and rRNA operon sequencing for species classification, we introduced random sequencing errors into the extracted sequences. Error rates were determined by referencing a 2022 study comparing the accuracy of Illumina and ONT technologies (24). 1D ONT MinION read error rates were applied to the rRNA operon sequences, while the average error rate of Illumina's read1 and read2 was used for the 16S rRNA sequences (Table 2). Errors were introduced through random positional mismatches, insertions, and deletions.

For each position, a random number between 0 and 1 was generated. If this number was lower than the error rate, an error was introduced. Mismatches involved replacing the original nucleotide with a random one. Implementation was carried out using BioPython SeqIO (25).

## Species classification within the genus

Two distinct methods were employed to classify species within the same genus: BLAST alignment score (26) and k-mer matching.

The BLAST-based approach used the sequences of the rRNA operon or 16S rRNA (including random errors) as the query, while the original sequences of the extracted regions served as the reference. Nucleotide BLAST was run with default options. Each sequence was classified into the species with the highest bitscore. In cases of ties, one species was randomly chosen for classification.

The k-mer matching method benchmarked the approach commonly used in microbiome data classification by Kraken (27). This method involves finding the number of exact matches of 31-mers and classifying the sequence to the species with the

**TABLE 2**  Error rates applied to simulate sequencing inaccuracies in rRNA operon and 16S rRNA sequences for species classification simulations[a]

| Marker region | Sequencing tool | Mismatch rate | Insertion rate | Deletion rate |
|---|---|---|---|---|
| rRNA operon | Nanopore | 0.0116 | 0.0081 | 0.0144 |
| 16S rRNA | Illumina | 0.0089 | 0.00045 | 0.00045 |
| 16S rRNA V3-V4 | Illumina | 0.0089 | 0.00045 | 0.00045 |

[a]The table outlines the mismatch, insertion, and deletion rates for the rRNA operon sequenced with Nanopore technology and the 16S rRNA (including its V3-V4 regions) sequenced with Illumina.

most 31-mer matches. Similar to the BLAST approach, ties were resolved by randomly choosing one species for classification.

In both methods, when multiple copies of the rRNA operon or 16S rRNA were present, we classified based on the copy with the highest score or the greatest number of matches. To determine the accuracy of species classification within each genus, we computed the proportion of correctly identified species among the samples, grouped by genus. Specifically, accuracy was calculated by comparing the actual species with the species identified through our classification methodology across different regions. For each genus, the accuracy rate was derived by dividing the number of correct species identifications by the total number of samples within that genus. The accuracy of each genus is assessed independently.

## Simulation of microbial community data

To evaluate the accuracy of species classification in community data, a simulation was run on community composition data. First, we used a Dirichlet distribution to randomly set proportions for the species in our study and made a mock proportion data. The alpha parameter of the Dirichlet distribution was itself derived from a geometric distribution with a probability of 0.01. Reads were initially distributed to match these true species proportions. Based on the likelihood of their classification through k-mer matching from the previous analysis, reads were then assigned to species. For example, if species A was correctly classified 90% of the time and misclassified as species B 10% of the time, a read intended for species A would have a 90% chance of being assigned to A and a 10% chance to B.

This process was applied to all reads. We conducted simulations for library sizes of 5,000, 10,000, 50,000, 100,000, 500,000, and 1,000,000 reads.

## Microbial community analysis and differential abundance analysis in real-world data

To further validate our findings using real-world gut microbiome data, we performed simulations with publicly available metagenomic data. We collected fecal metagenome sequencing data from healthy individuals and Crohn's disease patients to profile the gut microbiota. We downloaded the data from the SRA under project ID PRJEB15371 (28) and calculated the proportion of each species using MetaPhlAn 4.1 (29). Assuming the obtained proportions reflect reality, we predicted species proportions based on the classification accuracy of the rRNA operon, 16S rRNA, and 16S rRNA V3-V4 regions. This procedure mirrored our previous community composition simulation, again using a library size of 100,000 reads. Subsequently, we employed the R package Maaslin2 to conduct differential abundance analysis comparing healthy individuals and patients with Crohn's disease (30). The analysis utilized arcsine square root transformation (AST), total sum scaling (TSS) normalization, and a linear model. We considered findings with a $P$ value of less than 0.05 to be statistically significant.

## ACKNOWLEDGMENTS

We would like to express our gratitude to eGnome, Inc., for their support throughout the course of this research.

## AUTHOR AFFILIATIONS

[1]Interdisciplinary Program in Bioinformatics, Seoul National University, Seoul, Republic of Korea
[2]eGnome, Inc., Seoul, Republic of Korea
[3]Department of Agricultural Biotechnology, Research Institute of Agriculture and Life Sciences, Seoul National University, Seoul, Republic of Korea

## AUTHOR ORCIDs

Sohyoung Won ⓘ http://orcid.org/0000-0003-2039-6017
Heebal Kim ⓘ http://orcid.org/0000-0003-3064-1303

## ADDITIONAL FILES

The following material is available online.

### Supplemental Material

**Table S1 (Spectrum00931-24-s0001.xlsx).** The accessions of the genomes used for the study.
**Table S2 (Spectrum00931-24-s0002.xlsx).** The accuracy, precision, and sensitivity measured using different methods and regions per genus.

### Open Peer Review

**PEER REVIEW HISTORY (review-history.pdf).** An accounting of the reviewer comments and feedback.

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
