## [Reviewer comments · Microbiology Spectrum]

Microbiology Spectrum

rRNA Operon Improves Species-Level Classification of Bacteria and Microbial Community Analysis Compared to 16S rRNA

Sohyoung Won, Seoae Cho, and Heebal Kim

Corresponding Author(s): Heebal Kim, Seoul National University

Review Timeline:

Submission Date:	April 12, 2024
Editorial Decision:	May 9, 2024
Revision Received:	June 6, 2024
Editorial Decision:	June 20, 2024
Revision Received:	July 26, 2024
Accepted:	August 9, 2024

Editor: Yanbin Yin

Reviewer(s): The reviewers have opted to remain anonymous.

Transaction Report:

DOI: <https://doi.org/10.1128/spectrum.00931-24>

Re: Spectrum00931-24 (rRNA Operon Improves Species-Level Classification of Bacteria and Microbial Community Analysis Compared to 16S rRNA)

Dear Prof. Heebal Kim:

Thank you for the privilege of reviewing your work. Below you will find my comments, instructions from the Spectrum editorial office, and the reviewer comments.

The reviewer comments are generally positive. Some major concerns are raised by reviewer #1. Please consider the additional analyses they raised or otherwise respond why not relevant.

Revision Guidelines

Sincerely,
Yanbin Yin
Editor
Microbiology Spectrum

Reviewer #1 (Comments for the Author):

The paper, entitled 'rRNA Operon Improves Species-Level Classification of Bacteria and Microbial Community Analysis Compared to 16S rRNA,' aims to quantitatively compare and analyze the rRNA operon, the 16S rRNA, and its V3-V4 regions in accuracy for individual species identification and species-level microbial community analysis. There are many areas that need to

be addressed before it can be accepted for publication.

Major comments:

1. The results are not surprising and won't contribute significantly to what we already know. This is because: (1) The rRNA operon (16S, 23S, and 5S rRNAs) contains more genetic information compared to the 16S rRNA and the V3-V4 region; consequently, it can achieve greater accuracy in taxa identification. (2) A similar study was published previously in 2019 (titled: Taxonomic resolution of the ribosomal RNA operon in bacteria: implications for its use with long-read sequencing), but the authors didn't mention that study.
2. The authors didn't clearly mention the database they used for bacterial taxa identification.
3. The authors mentioned in the results that 'The average accuracy for BLAST-based classification using the rRNA operon reached 0.999, with a standard deviation of 0.005.' 'k-mer matching yielded comparable results. The average accuracy using the rRNA operon was 0.999, exceeding the 0.918 observed for the 16S rRNA and 0.693 for the V3-V4 regions.' The authors should provide a detailed description and data for how these accuracy rates were calculated. A supplementary table for the taxa information using the rRNA operon, the 16S rRNA, and its V3-V4 regions, as well as the organisms, should be provided.
4. The authors should also compare and analyze the accuracy of the other rRNA genes, 23S and 5S rRNAs, for species identification.
5. The authors are advised to repeat the analyses using another large dataset, the Genome taxonomy database (GTDB).
6. The authors only discussed the accuracy of using the rRNA operon as a marker for species identification compared to 16S rRNA. However, they should also discuss the feasibility of utilizing the rRNA operon in real microbiome studies. This includes addressing questions such as the universal primers that can be used for rRNA operon amplification and sequencing, as well as the bioinformatics tools suitable for analyzing microbiomes derived from rRNA operon sequences.
7. The authors used the study titled 'The trans-kingdom battle between donor and recipient gut microbiome influences fecal microbiota transplantation outcome' to compare microbial community and taxa analyses using the rRNA operon, the 16S rRNA, and its V3-V4 regions in so-called real-world data. This study utilized metagenomics sequencing. In microbiome analyses with metagenomics sequencing, multiple single-copy genes rather than the rRNA operon (16S, 23S, and 5S rRNAs) are widely employed for microbial community and taxa analyses. Therefore, the authors should consider using a study that employs both 16S sequence and rRNA operon sequence for microbiome analyses.

Reviewer #2 (Comments for the Author):

The authors compared accuracy of species-level classification between rRNA operon sequences, the full length 16S rRNA sequence and the 16S V3-V4 region using simulated data. The manuscript is clear, concise and well written and the conclusion from the various analyses strongly indicates that rRNA operon sequencing significantly outperforms the 16S rRNA and the 16S rRNA V3-V4 region for species classification.

Some comments:

- 1) Please include a rough estimation of the expected lengths of each sequenced region (rRNA operon vs 16S rRNA vs 16S rRNA V3-V4). This would emphasize the point on improved species classification with longer sequenced reads.
- 2) How was accuracy calculated? Since accuracy is the main metric here, it will be good to elaborate on how these calculations were made in the methods. Also, including other metrics such as precision and sensitivity would be a valuable indication of false positives/false negatives.
- 3) A lot of data has been generated for species classification within the 72 genera used in the analyses and this has been summarized in Figure 1. Can this data be provided in supplemental to show which genera are the least/most affected by each sequencing approach? This would be helpful in guiding readers when selecting their method of choice.
- 4) Figure 2 is missing a legend. Is it accurate that the relative abundance of the top ten totaled to 3%? This seems low but could be due to how the simulations were carried out?
- 5) The Bray Curtis distance results is striking and informative. It would be good to present this data in a figure/table.
- 6) Line 137: Please briefly describe the context of original paper whereby the FMT data was obtained. It will help the reader understand what "patient" and "donor" are referring to. It will also be helpful to mention that the data use for the actual analysis is simulated.
- 7) Line 146: Please clarify what groups are the differential abundance analysis comparing between (is it between donors and patients?)
- 8) Line 165: which two species?
- 9) Line 265: The header should read "Simulation of microbial community data"

10) Table 2: Does 16S rRNA refer to the full length gene (~1500bp) and is this possible with Illumina sequencing?

11) Another interesting point in the manuscript is the difference in performance between BLAST based and k-mer based classification methods. Is there a reason why both approaches were used initially but only the kmer based methods were used for the microbial community data?

12) Since the analysis described in the manuscript is based on publicly available and simulated data, it would be good if the authors can add a sentence or two surrounding the need to conduct actual sequencing experiments to confirm this (or if there is data in the literature that already supports this and can be cited).

Reviewer #1 (Comments for the Author):

The paper, entitled 'rRNA Operon Improves Species-Level Classification of Bacteria and Microbial Community Analysis Compared to 16S rRNA,' aims to quantitatively compare and analyze the rRNA operon, the 16S rRNA, and its V3-V4 regions in accuracy for individual species identification and species-level microbial community analysis. There are many areas that need to be addressed before it can be accepted for publication.

→ Thank you for your valuable opinion.

Major comments:

1. The results are not surprising and won't contribute significantly to what we already know. This is because: (1) The rRNA operon (16S, 23S, and 5S rRNAs) contains more genetic information compared to the 16S rRNA and the V3-V4 region; consequently, it can achieve greater accuracy in taxa identification. (2) A similar study was published previously in 2019 (titled: Taxonomic resolution of the ribosomal RNA operon in bacteria: implications for its use with long-read sequencing), but the authors didn't mention that study.

→ (1) While our findings might seem obvious, the significance lies in quantitatively analyzing the accuracy of species identification using the rRNA operon compared to the 16S rRNA and its V3-V4 region, and the impact of this accuracy on community analysis. We aimed to provide numerical evidence for known facts and to determine the extent of their influence. (2) The referenced paper analyzes in the context of a phylogenetic tree, which differs from the direction of our study. Nonetheless, as it addresses a similar topic, we have included mentions of this paper in the Introduction and Discussion sections.

2. The authors didn't clearly mention the database they used for bacterial taxa identification.

→ We followed GTDB's taxa identification. Accession IDs were matched with data from the GTDB database, and the matched species and genera were used. This information has been added to the Methods section.

3. The authors mentioned in the results that 'The average accuracy for BLAST-based classification using the rRNA operon reached 0.999, with a standard deviation of 0.005.' 'k-mer matching yielded comparable results. The average accuracy using the rRNA operon was 0.999, exceeding the 0.918 observed for the 16S rRNA and 0.693 for the V3-V4 regions.' The authors should provide a detailed description and data for how these accuracy rates were calculated. A supplementary table for the taxa information using the rRNA operon, the 16S rRNA, and its V3-V4 regions, as well as the organisms, should be provided.

→ In addition to the actual taxa information for each sample, Supplementary Table S1 presents the species classifications obtained using the rRNA operon, 16S rRNA, and its V3-V4 region. Additionally, the accuracy for each genus is provided in Supplementary Table S2. The Methods section has been updated with the following explanation.

“To determine the accuracy of species classification within each genus, we computed the proportion of correctly identified species among the samples, grouped by genus. Specifically, accuracy was calculated by comparing the actual species with the species identified through our classification methodology across different regions. For each genus, the accuracy rate was derived by dividing the number of correct species identifications by the total number of samples within that genus. Each genus's accuracy is thereby assessed independently.”

Additionally, during the creation of the Supplementary Tables, cases where the rRNA operon was not extracted were excluded, leading to a decrease in the number of genomes used for the final analysis. Consequently, the values in the results have slightly changed.

4. The authors should also compare and analyze the accuracy of the other rRNA genes, 23S and 5S rRNAs, for species identification.

→ We additionally analyzed the accuracy of 23S rRNA and 5S rRNA and added the results in the manuscript.

5. The authors are advised to repeat the analyses using another large dataset, the Genome taxonomy database (GTDB).

→ As of May 13, 2024, out of the 584,382 records in the GTDB database, only 775 (0.13%) were not included in the NCBI database. Since we utilized all available data from NCBI, we believe there is little need for additional analysis with the GTDB data. Our analysis may appear to use fewer data than GTDB because we excluded atypical genomes and metagenome-assembled genomes, using only complete genomes. This was necessary for accurate extraction of rRNA operon sequences. The details regarding data filtering have been added to the Methods section.

6. The authors only discussed the accuracy of using the rRNA operon as a marker for species identification compared to 16S rRNA. However, they should also discuss the feasibility of utilizing the rRNA operon in real microbiome studies. This includes addressing questions such as the universal primers that can be used for rRNA operon amplification and sequencing, as well as the bioinformatics tools suitable for analyzing microbiomes derived from rRNA operon sequences.

→ There are instances where the rRNA operon is used as a marker for species identification. By using the MIrROR database and pipeline developed by Seol et al. (2022), species identification with the rRNA operon can be applied (titled: Microbial Identification Using rRNA Operon Region: Database and Tool for Metataxonomics with Long-Read Sequence). The study also mentions the sequences of universal primers. Therefore, species identification using the rRNA operon is currently feasible. This information has been added to the Discussion section of the

manuscript. Additionally, there are examples of species-level analysis conducted using MlrROR (titled: Long-read MinION™ sequencing of 16S and 16S-ITS-23S rRNA genes provides species-level resolution of Lactobacillaceae in mixed communities).

7. The authors used the study titled 'The trans-kingdom battle between donor and recipient gut microbiome influences fecal microbiota transplantation outcome' to compare microbial community and taxa analyses using the rRNA operon, the 16S rRNA, and its V3-V4 regions in so-called real-world data. This study utilized metagenomics sequencing. In microbiome analyses with metagenomics sequencing, multiple single-copy genes rather than the rRNA operon (16S, 23S, and 5S rRNAs) are widely employed for microbial community and taxa analyses. Therefore, the authors should consider using a study that employs both 16S sequence and rRNA operon sequence for microbiome analyses.

→ While a study employing both 16S sequences and rRNA operon sequences for microbiome analyses would have provided a good comparison, we could not find such data. Although Olivier et al. (2023) created a mock community and sequenced the data, the actual proportions of the mock community were not mentioned, and only one genus was analyzed, so we did not use this data. Therefore, we had to use metagenomics data.

The analysis of this data was not an attempt to analyze metagenomics data with amplicon sequencing, but rather a second simulation based on the actual existing microbiome proportions. Since the proportion data generated using the Dirichlet distribution differs from the actual microbiome composition, we aimed to reference the actual microbiome composition from the literature to predict the expected outcomes when using the rRNA operon and 16S rRNA. We revised the manuscript to avoid confusion and added a note in the discussion that actual sequencing experiment results are needed to confirm our findings.

Reviewer #2 (Comments for the Author):

The authors compared accuracy of species-level classification between rRNA operon sequences, the full length 16S rRNA sequence and the 16S V3-V4 region using simulated data. The manuscript is clear, concise and well written and the conclusion from the various analyses strongly indicates that rRNA operon sequencing significantly outperforms the 16S rRNA and the 16S rRNA V3-V4 region for species classification.

➔ Thank you for your valuable opinion.

Some comments:

1) Please include a rough estimation of the expected lengths of each sequenced region (rRNA operon vs 16S rRNA vs 16S rRNA V3-V4). This would emphasize the point on improved species classification with longer sequenced reads.

➔ We calculated the average lengths of the regions used in the study and mentioned it in the Discussion section.

2) How was accuracy calculated? Since accuracy is the main metric here, it will be good to elaborate on how these calculations were made in the methods. Also, including other metrics such as precision and sensitivity would be a valuable indication of false positives/false negatives.

➔ We elaborated how the accuracy was calculated in the Methods section. We also obtained precision and sensitivity then added the results to the figures and the manuscript.

3) A lot of data has been generated for species classification within the 72 genera used in the analyses and this has been summarized in Figure 1. Can this data be provided in supplemental to show which

genera are the least/most affected by each sequencing approach? This would be helpful in guiding readers when selecting their method of choice.

→ The data used to create Figure 1 (accuracy for each genus) has been added to Supplementary Table 2. Genera that were most and least affected by the sequencing methods were discussed in the Discussion section. Additionally, during the creation of the Supplementary Tables, cases where the rRNA operon was not extracted were excluded, leading to a decrease in the number of genomes used for the final analysis, which resulted in slight changes in the values of the results.

4) Figure 2 is missing a legend. Is it accurate that the relative abundance of the top ten totaled to 3%? This seems low but could be due to how the simulations were carried out?

→ The legend (species names) were initially omitted as they did not hold scientific significance, but they have now been added. Random proportions were assigned to all species used in the study (2,026 species), resulting in even the top 10 abundant species having relatively low quantities. We have rechecked the values to ensure there are no anomalies. Recognizing that such a distribution might differ from typical microbiome compositions, we also presented results based on real-world proportions.

5) The Bray Curtis distance results is striking and informative. It would be good to present this data in a figure/table.

→ We added figures showing the distance analysis results. Additionally, we included more explanation about this analysis in the Results and Discussion sections. However, we believe that the Aitchison distance better matches our intentions and provides better visualization. Therefore, we changed the distance metric from Bray-Curtis distance to Aitchison distance.

6) Line 137: Please briefly describe the context of original paper whereby the FMT data was obtained. It will help the reader understand what "patient" and "donor" are referring to. It will also be helpful to mention that the data use for the actual analysis is simulated.

→ We added some descriptions about the data in the Results section and explained what the "patient" and "donor" groups refer to.

7) Line 146: Please clarify what groups are the differential abundance analysis comparing between (is it between donors and patients?)

→ We compared healthy donors and patients before FMT. This information was also added while explaining the data in section 6.

8) Line 165: which two species?

→ The species were *Staphylococcus warneri* A and *Listeria monocytogenes* B. However, after reanalyzing the data, no differing species were found, so this information was removed.

9) Line 265: The header should read "Simulation of microbial community data"

→ We modified it.

10) Table 2: Does 16S rRNA refer to the full length gene (~1500bp) and is this possible with Illumina sequencing?

→ 16S rRNA refers to the full-length gene (~1500bp), and using synthetic long-read technology, it can be sequenced with Illumina. The paper titled "The effect of taxonomic classification by

full-length 16S rRNA sequencing with a synthetic long-read technology" demonstrated the sequencing and analysis of full-length 16S rRNA using Illumina sequencing.

11) Another interesting point in the manuscript is the difference in performance between BLAST based and k-mer based classification methods. Is there a reason why both approaches were used initially but only the kmer based methods were used for the microbial community data?

→ Including results from both methods would result in an excess of similar findings, so we presented the community analysis results using only one method. Since the k-mer based method is used in widely adopted community analysis tools like Kraken, we based our analysis on the results from this method. We anticipate that using the BLAST method would not significantly alter the results.

12) Since the analysis described in the manuscript is based on publicly available and simulated data, it would be good if the authors can add a sentence or two surrounding the need to conduct actual sequencing experiments to confirm this (or if there is data in the literature that already supports this and can be cited).

→ It would be beneficial to include this information in the Discussion section, so we added some sentence about this. However, we could not find related literature.

Re: Spectrum00931-24R1 (rRNA Operon Improves Species-Level Classification of Bacteria and Microbial Community Analysis Compared to 16S rRNA)

Dear Prof. Heebal Kim:

Thank you for the privilege of reviewing your work. Below you will find my comments, instructions from the Spectrum editorial office, and the reviewer comments.

Please return the manuscript within 30 days; if you cannot complete the modification within this time period, please contact me. If you do not wish to modify the manuscript and prefer to submit it to another journal, notify me immediately so that the manuscript may be formally withdrawn from consideration by Spectrum.

Revision Guidelines

Sincerely,
Yanbin Yin
Editor
Microbiology Spectrum

Reviewer #1 (Comments for the Author):

The authors have addressed most of my comments. However, there are still a few areas that need attention before it can be accepted for publication.

Major comments:

1. In the study of microbial community composition and differential abundance in human gut microbiome data, it is still unclear how the reference species were classified. The authors assumed that the reference species in the mentioned paper were correct but did not specify how these species were identified. It is advised that the authors reanalyze the metagenomics data using MetaPhlAn 4.0, given that the methods used in the 2020 paper may be outdated.

Minor comments:

1. In line 194, the reference should be put after FMT study.
2. In line 202, the statement 'The average and standard deviation were 0.210 and 0.166, respectively, with the 5S rRNA' is confusing and should be rewritten to avoid confusion.
3. In lines 204-206, the authors stated that 'Conversely, while the 23S rRNA had a high average correlation, it could drop as low as 0.651, indicating significant variability in prediction accuracy among samples.' The authors should be more specific about the conditions under which it could drop as low as 0.651.
4. In line 226, the species name '*Bacillus safensis*' should be italicized. The authors are advised to check all species and genus in the manuscript.

Reviewer #2 (Comments for the Author):

Thanks for addressing the previous comments thoroughly. The message whereby the rRNA operon outperforms other marker genes is clear and strongly supported by the various analyses.

One minor comment:

The sentence "k-mer matching yielded comparable results" is repeated twice in lines 86 and 87.

Reviewer #1 (Comments for the Author):

The authors have addressed most of my comments. However, there are still a few areas that need attention before it can be accepted for publication.

→ We appreciate your comments.

Major comments:

1. In the study of microbial community composition and differential abundance in human gut microbiome data, it is still unclear how the reference species were classified. The authors assumed that the reference species in the mentioned paper were correct but did not specify how these species were identified. It is advised that the authors reanalyze the metagenomics data using MetaPhlAn 4.0, given that the methods used in the 2020 paper may be outdated.

→ We analyzed the previous reference data (FMT data) using MetaPhlAn 4.1; however, this analysis yielded only a few species. We thought that this would be unsuitable for further analyses and subsequently obtained a different metagenomics dataset pertaining to Crohn's disease (PRJEB15371). We downloaded the raw sequence data from the SRA and profiled the species proportions using MetaPhlAn 4.1 with default settings. We then assumed these results to represent the true proportions and proceeded with our analyses accordingly. Consequently, we revised the figures and the manuscript to reflect these updated results.

Minor comments:

1. In line 194, the reference should be put after FMT study.

→ We put the reference after the sentence mentioning the study (Crohn's disease in the revised version).

2. In line 202, the statement 'The average and standard deviation were 0.210 and 0.166, respectively, with the 5S rRNA' is confusing and should be rewritten to avoid confusion.

→ We modified the sentence to separately mention average and standard deviation.

3. In lines 204-206, the authors stated that 'Conversely, while the 23S rRNA had a high average correlation, it could drop as low as 0.651, indicating significant variability in prediction accuracy among samples.' The authors should be more specific about the conditions under which it could drop as low as 0.651.

→ We clarified the sentence by explaining that the accuracy dropped in one patient's sample.

4. In line 226, the species name '*Bacillus safensis*' should be italicized. The authors are advised to check all species and genus in the manuscript.

→ The mentioned sentence was removed during the revision, and I verified that species and genus names are italicized in the remaining sections.

Reviewer #2 (Comments for the Author):

Thanks for addressing the previous comments thoroughly. The message whereby the rRNA operon outperforms other marker genes is clear and strongly supported by the various analyses.

→ We appreciate your comments.

One minor comment:

The sentence "k-mer matching yielded comparable results" is repeated twice in lines 86 and 87.

→ We removed the repeated sentences.

Re: Spectrum00931-24R2 (rRNA Operon Improves Species-Level Classification of Bacteria and Microbial Community Analysis Compared to 16S rRNA)

Dear Prof. Heebal Kim:

Your manuscript has been accepted, and I am forwarding it to the ASM production staff for publication. Your paper will first be checked to make sure all elements meet the technical requirements. ASM staff will contact you if anything needs to be revised before copyediting and production can begin. Otherwise, you will be notified when your proofs are ready to be viewed.

Sincerely,
Yanbin Yin
Editor
Microbiology Spectrum